# Protective Mechanisms of Polyphenol-Enriched Blueberry Preparation in Preventing Inflammation in the Skin against UVB-Induced Damage in an Animal Model

**DOI:** 10.3390/antiox13010025

**Published:** 2023-12-21

**Authors:** Nawal Alsadi, Hamed Yasavoli-Sharahi, Rudolf Mueller, Cyrille Cuenin, Felicia Chung, Zdenko Herceg, Chantal Matar

**Affiliations:** 1Cellular and Molecular Medicine Department, Faculty of Medicine, University of Ottawa, Ottawa, ON K1H 8M5, Canada; nalsa068@uottawa.ca (N.A.); hyasa068@uottawa.ca (H.Y.-S.); 2Pathology and Laboratory Medicine Department, Faculty of Medicine, University of Ottawa, Ottawa, ON K1H 8M5, Canada; rudi.mueller1@gmail.com; 3Epigenomics and Mechanisms Branch, International Agency for Research on Cancer (IARC), 25 Av. Tony Garnier, 69007 Lyon, France; cueninc@iarc.fr (C.C.); feliciacfl@sunway.edu.my (F.C.); hercegz@iarc.fr (Z.H.); 4Department of Medical Sciences, School of Medical and Life Sciences, Sunway University, Jalan University, Bandar Sunway, Subang Jaya 47500, Malaysia; 5School of Nutrition, Faculty of Health Sciences, University of Ottawa, Ottawa, ON K1H 8M5, Canada

**Keywords:** ultraviolet radiation (UVB), polyphenols, PEBP, miRNA, nuclear factor kappa-light-chain-enhancer of activated B cells (NF-κB)

## Abstract

UVB significantly impacts the occurrence of cutaneous disorders, ranging from inflammatory to neoplastic diseases. Polyphenols derived from plants have been found to exhibit photoprotective effects against various factors that contribute to skin cancer. During the fermentation of the polyphenol-enriched blueberry preparation (PEBP), small oligomers of polyphenols were released, thus enhancing their photoprotective effects. This study aimed to investigate the protective effects of PEBP on UVB-induced skin inflammation. Topical preparations of polyphenols were applied to the skin of dorsally shaved mice. Mice were subsequently exposed to UVB and were sacrificed 90 min after UVB exposure. This study revealed that pretreatment with PEBP significantly inhibited UVB-induced recruitment of mast and neutrophil cells and prevented the loss of skin thickness. Furthermore, the findings show that PEBP treatment resulted in the downregulation of miR-210, 146a, and 155 and the upregulation of miR-200c and miR-205 compared to the UVB-irradiated mice. Additionally, PEBP was found to reduce the expression of IL-6, IL-1β, and TNFα, inhibiting COX-2 and increasing IL-10 after UVB exposure. Moreover, DNA methylation analysis indicated that PEBP might potentially reduce the activation of inflammation-related pathways such as MAPK, Wnt, Notch, and PI3K-AKT signaling. Our finding suggests that topical application of PEBP treatment may effectively prevent UVB-induced skin damage by inhibiting inflammation.

## 1. Introduction

The skin is considered the largest organ in the human body. It acts as a barrier, protecting organs from harmful environmental stimuli. It is one of the body’s most complex and reactive organs and contains the epidermis and the vascularized dermis [1]. Ultraviolet radiation (UV) is a major environmental carcinogen and the primary cause of cancer and skin pathologies such as erythema, inflammation, and degenerative aging changes [2,3]. UVB, in particular, is the most damaging form of UV radiation to human skin [4,5]. UVB can induce the formation of reactive oxygen species (ROS) that increase oxidative stress, induce the infiltration of neutrophils into the dermis, the production of pro-inflammatory mediators, and, eventually, skin cancer [2,6,7,8].

Antioxidant supplementation by topical application is recommended as an effective strategy to mitigate the adverse effects of UVB radiation on the skin [9,10,11]. Compounds with photoprotective activity, such as polyphenols derived from natural sources, have gained significant attention in recent years for their potential to prevent UVB-induced skin cancer and inflammation [10,12]. For instance, polyphenols from blueberries, including catechin, epicatechin, and oligomeric proanthocyanidins, have been used as potential treatments to protect the skin from the damaging effects of UV radiation [8,11,13,14]. Moreover, the fermentation of blueberries results in the release of small polyphenols with enhanced bioavailability. The biotransformation process yields a polyphenol-enriched blueberry preparation (PEBP), which contains gallic acid (GA), protocatechuic acid (PCA), and catechin (Cat) [11,14,15]. In a previous study in our lab, an oligomeric mixture of polyphenols (OMP) was designed to mimic the fermentation process by utilizing a balanced polyphenol blend of GA, PCA, and Cat [13,14]. These compounds have demonstrated possible UV-protective and antimutagenic, anti-diabetic, anti-inflammatory, anti-cancer, antioxidant, and immunomodulatory properties [16,17]. Studies have shown that topical pretreatment with certain polyphenolic compounds prior to UVB exposure significantly reduces UVB-induced inflammation and skin tumorigenesis in mouse models [15,18,19].

The molecular mechanisms underlying the changes in the skin induced by chronic UVB exposure remain unclear. However, studies have implicated epigenomic alterations, such as miRNAs and DNA methylation, which are indirectly involved in UVB-regulated apoptosis, inflammation, and cell cycle control [10,20]. MicroRNAs play a crucial role in regulating skin homeostasis and mitigating or exacerbating damage caused by UVB radiation [20]. These small, non-coding RNAs can target specific mRNAs and induce their degradation, thereby inhibiting protein translation [20,21]. Numerous miRNAs have been shown to regulate processes such as DNA damage, photoaging, cell survival, carcinogenesis, and pigmentation, and their expression profiles differ following exposure to UVB radiation [16,22]. UVB irradiation decreases the expression level of miR-205 and miR-200c, which are essential for motility and cell-cell adhesion [23], while increased miR-155, miR-210, and miR-146a lead to exceptional responsiveness to many inflammatory stimuli [24,25,26,27].

UV irradiation is a well-known trigger for increasing nuclear factor kappa light chain enhancer of activated B (NF-κB) transcriptional activity, subsequently leading to a chronic inflammatory signal [28,29]. NF-κB is a key mediator of cellular inflammatory processes that induce pro-inflammatory cytokine expressions such as interleukin-1β (IL-1β), interleukin-6 (IL-6), and tumor necrosis factor α (TNF-α) [29,30]. Activation of NF-κB pathways also inhibits the activation of anti-inflammatory cytokines such as interleukin-10 (IL-10) [29,31]. These cytokines play crucial roles in inflammatory skin diseases and skin cancer. The changes in cytokine production are believed to occur through the activation of the NF-κB pathway [30,32].

DNA methylation plays a critical role in the epigenome as a fundamental mechanism for controlling gene expression and the regulation of cellular pathways [33,34,35]. DNA methylation has been shown to be involved in the regulation of inflammation, which encompasses cancer development. Therefore, modulation of methylation is a promising strategy for cancer prevention and treatment [33,34,35]. Notably, the impact of short-term exposure to UVB radiation results in changes in the immune response, including the inflammation and DNA damage pathways [36], which might influence the activation or repression of specific genes involved in multiple related pathways [16,37]. These modifications can potentially activate oncogenes or suppress tumor suppressor genes, ultimately contributing to skin cancer initiation [33,38,39]. Nevertheless, despite these findings, further exploration is essential to enhance our understanding of the epigenetic changes triggered by acute UV irradiation and their wide-ranging implications in the captivating field of skin photobiology.

Numerous studies have demonstrated the beneficial effects of polyphenol products on human skin, including photoprotection and improvement of physiological parameters [40,41]. However, the existing research on the protective effects of fermented blueberries against skin damage caused by UVB radiation remains limited. Therefore, we investigated the photoprotective effects of two specific compounds, PEBP and OMP, against short-term UVB exposure. Specifically, our study focused on examining their ability to prevent the inflammatory increase in cytokines and activation of the NF-κB response in skin mice, which are critical indicators of UV-induced damage. Additionally, we explored the potential involvement of miRNAs and methylation mechanisms in this process. Therefore, using natural bioactive compounds in cosmetics and medical research to prevent UV-induced skin damage has significant chemoprevention potential.

## 2. Materials and Methods

### 2.1. Natural Products

Fresh and untreated, fully matured wild blueberries (*Vaccinium angustifolium* Ait.) were carefully harvested from designated areas in the Atlantic region. Subsequently, the blueberries underwent centrifugation at 500× *g* for 10 min using an IEC Centra MP4R centrifuge (International Equipment Company, Needham Heights, MA, USA). This step aimed to eliminate fruit skin and non-homogenized particles. To ensure purity, the resulting juice underwent sterilization via filtration through a 0.22 µm Express Millipore filter apparatus from Millipore, Etobicoke, ON, Canada.

Culturing SV-53 bacteria followed established procedures as previously described [42]. Details regarding the characterization of blueberry and polyphenol-enriched blueberry preparations can be found in previous studies [42,43]. This compound will be described as the “Polyphenols-Enriched Blueberry Preparation (PEBP)”.

The ultra-performance liquid chromatography-quadrupole time-of-flight mass spectrometry (UPLC-MS-QTOF) analysis and fractionation of the fermented blueberry extract, also known as PEBP, revealed that fractions rich in bioactive compounds like gallic acid (GA), protocatechuic acid (PCA), and catechins (Cat) effectively contribute to maintaining glucose homeostasis. Standards with over 95% purity for the major compounds found in blueberries, including PCA, GA, and Cat, were procured commercially from Sigma-Aldrich (St. Louis, MO, USA) [14]. Throughout this manuscript, these compounds will be denoted as the “Oligomeric Mixture of Polyphenols (OMP)”, emphasizing their significance.

### 2.2. Animal

BALB/c female mice, 6–8 weeks of age, were housed in the animal facility at the University of Ottawa. The mice were housed three per cage under constant humidity and temperature with 12 h light/dark cycles. They were allowed access to water and standard mouse feed ad libitum and were monitored daily. The animal protocol for this study was approved by the Institutional Animal Care and Use Committee of the University of Ottawa.

### 2.3. Preparation of Polyphenols Containing Cream

For the following in vivo studies, a water-in-oil (W/O) cream containing different polyphenol compounds was formulated. The cream base was formulated by mixing 50 g of an aqueous solution comprised of glycerine (0.05% *w*/*w*) and Tween^®^-60 (0.10% *w*/*w*) with 25 g of Vaseline and 10 g of cetylstearil alcohol that had been melted separately at 70 °C under constant agitation. The resultant emulsion was then cooled to room temperature under constant agitation. Experimental formulations were then prepared by adding either the vehicle cream, water (0.3%), or 0.3% of non-fermented blueberry juice (NBJ), polyphenol-enriched blueberry preparation (PEBP), or an oligomerized mixture of polyphenols (OMP). Creams containing NBJ, PEBP, or OMP were stored at 4 °C in the dark. The pH of the creams was adjusted to pH 7.0 with a sodium hydroxide solution.

### 2.4. UV Irradiation

Before irradiation, BALB/c female mice (6–8-week-old) (six mice in each group) were shaved using an electric razor. The skin was sterilized by chlorhexidine and 70% ethanol washes. Mice were covered with a pre-designed shield. Each mouse is treated with 1.2 mg of each cream and then exposed to UVB irradiation (290–320 nm) for 90 min. After 90 min of exposure to UVB light, all animals were sacrificed. Dorsal skin (approximately 2 × 2 cm^2^ pieces) was excised from each mouse at the exposure sites as well as the control sites (areas that were not exposed to UVB light). Each skin sample was cut into three pieces and used for subsequent analysis. To maintain a similar treatment protocol and effects, non-UVB-exposed control groups of mice were also treated with the same doses of NBJ, PEBP, or OMP. The skin tissues were collected and pooled from each mouse in each treatment group. 

### 2.5. Histopathology

To evaluate the effect of NBJ, PEBP, or OMP on UV-induced changes in skin morphology, the skin tissues were dissected from the respective groups and fixed with 10% formalin for at least 48 h at room temperature. After fixation, the tissues were processed for paraffin embedding. Subsequently, 4 mm-thick tissue sections were cut from paraffin blocks and stained with haematoxylin and eosin (H&E) as per standard protocol. The histopathological analysis was performed by a board-certified pathologist, and the images were taken using light microscopy. Image J software 1.54 (National Institutes of Health, Bethesda, MD, USA) was used to estimate the thickness of the epidermal and dermal layers.

### 2.6. RNA Extraction and Real-Time Quantitative PCR (RT-qPCR)

Total RNA from the skin samples was extracted using a Trizol reagent (Life Technologies, Carlsbad, CA, USA) according to the manufacturer’s protocol. MiRNAs were extracted using a miRNeasy kit (Qiagen, Toronto, ON, Canada) and quantified using a NanoDrop spectrophotometer (NanoDrop ND1000; Thermo Fisher Scientific, Waltham, MA, USA). cDNA was synthesized from 1 µg of RNA using the iScript cDNA synthesis kit (Bio-Rad, Hercules, CA, USA) according to the manufacturer’s instructions. The cDNA was then diluted 1:60 in nuclease-free water, and qPCR was performed using miRCURY™ SYBR Green Master Mix (Qiagen, Toronto, ON, Canada). cDNA was amplified by real-time PCR with a Bio-Rad MyiQ thermocycler and SYBR Green detection system (Bio-Rad, Hercules, CA, USA). The standard PCR conditions were 95 °C for 10 min and then 40 cycles at 95 °C for 30 s, 60 °C for 30 s, and 72 °C for 30 s in a CFX96 machine (Bio-Rad, Mississauga, ON, Canada). U6 was used as an internal control for normalization in each sample (Applied Biosystems, Burlington, ON, Canada). For miRNA analysis, the calculations for determining the relative level of gene expression were made using the cycle threshold (Ct) method. The mean Ct values from duplicate measurements were used to calculate the expression of the target gene with normalization to a housekeeping gene used as an internal control and using the equation: relative quantity (RQ) = 2^−ΔΔCT^ algorithm.

### 2.7. Western Blotting

Total proteins from skin tissues were extracted and homogenized in 400 μL Pierce^®^ RIPA buffer (Sigma-Aldrich, St. Louis, MO, USA) supplemented with phosphatase inhibitor cocktails (100×) (Thermo Scientific, Waltham, MA, USA, catalog number: 78440). Protein concentrations were measured using a BCA protein assay kit (no. 23227; Thermo Fisher Scientific, Waltham, MA, USA). Protein samples (50 μg) were separated by 12% SDS-PAGE gels, then transferred to polyvinylidene fluoride membranes (PVDF) (Invitrogen, Burlington, ON, Canada). After blocking with 5% fat-free milk, membranes were incubated with primary antibodies (1:500) NF-κB (Cell Signaling Tech. Inc., Danvers, MA, USA), COX-2 (Santa Cruz, Dallas, TX, USA), IL-1β (Abcam, Cambridge, MA, USA), and TNF-α (Proteintech, San Diego, CA, USA) and incubated overnight at 4 °C, followed by incubation with appropriate secondary antibodies (1:5000) for 1 h (Jackson Immuno Research Laboratories, West Grove, PA, USA). The membrane was washed in TBS-T and treated with a chemiluminescence reagent ECL detection kit (Bio-Rad, Mississauga, ON, Canada) according to the manufacturer’s protocol. Next, transferred protein bands were visualized and analyzed using the chemiluminescence imaging system (Bio-Rad, Mississauga, ON, Canada).

### 2.8. Tissue Immunohistochemistry

The amount of 4 μm-thick formalin-fixed skin sections was deparaffinized, hydrated through xylene and graded alcohols, and washed three times. Antigen retrieval was done with citrate buffer (pH 6.0), and the tissue sections were incubated in peroxidase block for 10 min. After two successive washes with TBS-Tween 20, sections were incubated in protein-blocking serum (Dako Diagnostics, Missisauga, ON, Canada) for 30 min and incubated with the various primary antibodies IL-10 (1:50) and IL-6 (1:50) overnight. Two successive washes were done, and the sections were incubated with the secondary antibody-conjugated fluorophores IL-10 (Alex 555) and IL-6 (Alexa 488) for 1 h. Then, slides were washed two times with TBS-Tween 20 buffer. Then the DAPI working reagent was prepared immediately, and a few drops were dropped on the slides for 3 min. After rinsing slides in distilled water, they were mounted using ProLong Gold and VECTASHIELD antifade mounting medium (Thermo Fisher Scientific, Toronto, ON, Canada) and visualized using a Zeiss LSM 880 AxioObserver Z1 Confocal Microscope. (Leica, Wetzel, Germany). For a negative control, sections were treated with a primary antibody dilution solution. Quantification was generated from eight fields of view from a representative experiment and analyzed by ImageJ 1.54.

### 2.9. DNA Extraction and DNA Methylation Analysis

Approximately 15 to 20 mg of samples were homogenized using an electrical homogenizer (Bead Mill 24, Fisher Scientific, Waltham, MA, USA) in tubes containing 500 μL of cell lysis buffer and 1.5 microliters of proteinase K. The extraction of tissue DNA was carried out using the Gentra Puregene Tissue Kit (Qiagen, Toronto, ON, Canada), following the manufacturer’s guidelines. The obtained DNA was then diluted with a rehydration solution to achieve a final concentration of 20 ng/μL and stored at a temperature of −20 °C. The concentration of the extracted DNA was measured using the Qubit 4 system (Thermo Fisher Scientific, Toronto, ON, Canada).

DNA methylation analysis was performed as previously reported [44]. Briefly, 500 ng of the extracted DNA underwent a bisulphite conversion process using the EZ DNA Methylation kit (Zymo Research, Irvine, CA, USA). Subsequently, 250 ng of the bisulphite-modified DNA were examined using the Infinium Mouse Methylation BeadChip arrays (Illumina Inc., San Diego, CA, USA). This enabled the simultaneous assessment of DNA methylation at more than 285,000 CpG sites. The methylome-wide data were then processed using the methylkey pipeline, a software tool developed by the Epigenomics and Mechanisms Branch at the International Agency for Research on Cancer (https://github.com/IARCbioinfo/methylkey, accessed on 23 January 2023). This pipeline involved various steps, such as raw data preprocessing, quality control measures, and data normalization using Noob normalization through the SeSAMe package 3.9. To compare different groups, intergroup comparisons were carried out using linear regression analysis with the assistance of the limma R package 4.2.3. Regional analysis was conducted to identify regions with differential methylation patterns using the DMRcate package 3.18.

### 2.10. Bioinformatics Analysis

Pathway significance enrichment analysis was determined using the Enrichr package KEGG enrichment analysis in R (https://maayanlab.cloud/Enrichr/, accessed on 29 March 2023). This package was used to identify pathways significantly enriched in candidate genes compared with the whole genome background. Pathways with a *p*-value ≤ 0.05 were defined as pathways enriched considerably in UVB.

### 2.11. Statistical Analysis

Error bars represent the mean ± SEM from at least three separate experiments. The statistical analysis was performed using GraphPad Prism 8.0 software (GraphPad Software Inc., San Diego, CA, USA). A one-way analysis of variance (ANOVA) followed by a post hoc test was used to assess differences between more than two groups, with *p* < 0.05 considered statistically significant. Statistically significant results were defined as follows: * *p* < 0.05; ** *p* < 0.01; *** *p* < 0.001; **** *p* < 0.0001. Differentially methylated genes were defined with a false discovery rate (FDR)-adjusted *p*-value cutoff of ≤0.05 and a group mean difference of ≥3%. For pathway visualization, KEGG pathway enrichment analysis was performed using Enrichr.

## 3. Results

### 3.1. PEBP Decreases the Mast Cell, Neutrophil Cells Count and Prevented Loss of the Skin Thickness in BALB/c Mice Skin following a Short-Term UVB Exposure

Histological examinations of the exposed BALB/c skin samples revealed that UVB-induced mast cell infiltration was significantly reduced in samples where the mice had been pretreated with formulations containing PEBP or OMP compared to exposed skin that had been pretreated with control cream (Figure 1A,B). Similarly, the neutrophil count was significantly increased in UVB-exposed dermis samples compared to non-irradiated controls. PEBP, OMP, and NBJ-treated mice had fewer neutrophils than UVB-treated mice (Figure 1C). Epidermal thickness was also significantly higher in UVB mice compared to non-irradiated controls. This increase in thickness was abrogated by pretreatment with the tested formulations, NBJ, PEBP, and OMP, when compared to UVB-irradiated mice pretreated with control cream (Figure 1D).

### 3.2. PEBP and OMP Modulate miRNA Expression in Response to UVB Radiation, Upregulating miR-200c and miR-205 while Downregulating miR-210, miR-155, and miR-146a

To investigate the impact of PEBP and OMP on the regulation of miRNAs associated with skin inflammation following short-term UVB exposure, we analyzed the expression of miR-210, miR-200c, miR-146a, miR-155, and miR-205 in skin tissues using RT-qPCR. Our results demonstrated that the group treated with PEBP and OMP exhibited a significant decrease in miR-155 (Figure 2A), miR-210 (Figure 2B), and miR-146a (Figure 2C) expression compared to the UVB control group. Conversely, miR-205 (Figure 2D) and miR-200c (Figure 2E) expressions were upregulated after treatment with PEBP, OMP, and NBJ compared to the UVB control group. Furthermore, we observed that PEBP was more effective than OMP or NBJ when treating the skin before UVB exposure. These findings suggest that PEBP might have the ability to counteract UVB-induced changes in miRNA expression (Figure 2).

### 3.3. After Short-Term UVB Radiation, PEBP and OMP Treatment Decreased the Expression of NF-κB Activation in BALB/c Mice Skin

To further analyze the photoprotective mechanisms, the role of treatment on NF-κB activation in UVB-irradiated BALB/c mouse skin was studied. The activation of NF-κB p65 plays a crucial role in enhancing downstream target gene expression in the skin, particularly in response to inflammatory cytokines such as TNF-α and IL-1β [45]. Correspondingly, we observed that UVB exposure induced NF-κB activation expression, which was accompanied by upregulation of TNF-α, IL-1β, and COX-2. Skin tissue from mice pretreated with PEBP, OMP, and NBJ displayed significantly lower levels of NF-κB p65, TNF-α, IL-1β, and COX-2 expression, indicating an inhibition of NF-κB activity (Figure 3). These findings suggest that PEBP, OMP, and NBJ inhibit NF-κB (p65) expression by reducing COX-2 and pro-inflammatory markers such as TNF-α and IL-1β, thereby preventing inflammatory damage to the skin.

### 3.4. PEPB and OMP Modulate the Expression of Pro-Inflammatory Cytokines (IL-6) and Anti-Inflammatory Cytokines (IL-10) in UVB-Irradiated Skin Samples

The above results were corroborated by observations that skin tissue samples from mice pre-treated with PEBP and OMP showed inflammatory biomarkers following short-term UVB exposure to their skin. We performed immunofluorescence staining to evaluate the levels of pro-inflammatory cytokine IL-6 and anti-inflammatory cytokine IL-10 expression in all treatment groups (Figure 4). Our results showed that the skin tissue harvested from mice in the UVB + PEBP and UVB + OMP groups displayed significantly lower levels of IL-6 compared to the vehicle-treated + UVB group. On the other hand, IL-10 expression was significantly lower in the UVB + PEBP and UVB + OMP groups compared to the vehicle-treated + UVB groups (Figure 4). These findings suggest that PEBP and OMP have the potential to modulate the expression of pro-inflammatory and anti-inflammatory cytokines in UVB-irradiated skin samples.

### 3.5. Topical Application of PEBP Modulates DNA Methylation Patterns in Mice Skin following Short-Term UVB Exposure

DNA methylation is a crucial epigenetic process responsible for silencing DNA [46]. The methylation state of particular gene regions, such as the regions 1–5 Kb upstream transcriptional start points, the promoter, and the 5′UTR, can influence this silencing effect [34,44]. DNA methylome-wide analysis revealed that there was extensive differential methylation (1146 differentially methylated regions (DMRs)) with an FDR-adjusted *p*-value cutoff of ≤0.05 and a group mean difference of ≥3% (Figure 5A) when comparing the UVB-exposed and non-exposed groups. The majority of these DMRs were hypomethylated (869 DMRs, 75.82%), while 277 DMRs (24.17%) were hypermethylated in the UVB+ group relative to the UVB- group. Relative to the overall probe distribution on the array, the DMRs were enriched for intronic regions and depleted for regions 1–5 Kb upstream of the TSS and exonic regions (Figure 5B). In terms of their relationship to the closest CpG islands, the DMRs were enriched for open sea regions and depleted for CpG shores and CpG islands (Figure 5B).

When comparing UVB vs. UVB + PEBP groups, we observed 850 DMRs, 679 (79.89%) of which were hypomethylated, and 171 (20.12%) of which were hypermethylated (Figure 5A). The significantly hypomethylated regions were enriched for intronic regions, while the significantly hypermethylated regions were enriched for exonic regions, relative to the overall distribution of the array. Significantly hypomethylated regions were enriched for CpG shores and depleted for open sea regions, while significantly hypermethylated regions were enriched for open sea and exonic regions. Interestingly, no significant differences in DNA methylation patterns were observed between samples from the −UVB control groups and the PEBP groups. Furthermore, it is essential to note that no significant differences were found between the UVB group, and the groups treated with NBJ and OMP. As a result, the only groups that were significantly different in terms of their DNA methylation patterns were comparisons between the UVB control group and the UVB radiation group, and the UVB group compared to the UVB/PEBP. These results suggest that UVB irradiation and PEBP effects can alter the DNA methylation profiles of mouse epidermal cells.

Additionally, KEGG pathway enrichment chart analyses revealed 24 significantly enriched pathways (*p* < 0.05, Figure 5C). These pathways are constituted of differentially methylated genes implicated in cancer-related pathways, cell growth, inflammation, and death-related pathways, such as the phosphatidylinositol 3-kinase (PI3K)/protein kinase B (AKT), Mitogen-activated protein kinase (MAPK), the transforming growth factor beta (TGFb), Wnt pathway, and Notch signaling pathway. Moreover, adhesion junction, CGMP-PKG signaling pathway, and Hippo signaling pathway (*p* < 0.05). These results suggest that PEBP treatment impacts multiple pathways involved in regulating inflammatory signaling pathways.

The effect of DNA methylation on the key pathways was further analyzed by selecting the most significant genes affected by UVB exposure. We observed that the pretreatment of PEBP partially abrogated the effects of UVB exposure in skin tissue (Figure 5D). The findings revealed that PEBP treatment resulted in significant hypermethylation of genes such as Thousand and one amino acid (TAO) kinases1 (*Taok1*), Mitogen-Activated Protein Kinase Kinase Kinase Kinase 4 (*Map4k4*), Tyrosine 3-Monooxygenase/Tryptophan 5-Monooxygenase Activation Protein Zeta (*Ywhaz*), Suppressor of Mothers against Decapentaplegic 6 (*Smad6*), Interferon Alpha 2 (*Ifna2*), and Interferon Alpha k (*Ifnk*), while genes such as Runt-related transcription factor 1 (*Runx1*) and Forkhead Box P1 (*Foxp1*) exhibited hypomethylation following PEBP treatment (Figure 5D). Therefore, our results suggest that PEBP significantly impacts the UV-induced hypomethylation or hypermethylation patterns of DNA in vivo.

## 4. Discussion

UVB irradiation is known to cause skin aging, inflammation, and DNA damage [10]. As continued exposure to UVB irradiation remains an ongoing health hazard, there remains a need to identify protective mechanisms against the detrimental health effects of UVB radiation using UV screens or other direct or indirect approaches [47]. UVB radiation can induce inflammatory mediators such as COX-2 and pro-inflammatory cytokines, including TNF-α, IL-6, and IL-1β, which induce the expression of NF-κB, which has been shown in inflammatory skin diseases [28,48]. Recently, it has been demonstrated that several naturally occurring active compounds can protect skin from UVB [10,49,50] by inducing protective cellular mechanisms related to reducing oxidative stress, DNA damage, and inflammation [51]. Increased consumption of antioxidant substances, including flavones, protocatechuic acid, catechins, and polysaccharides, was identified as a potential strategy for protecting against the damaging effects of excessive UVB radiation [10,19].

There is increasing evidence from in vitro and in vivo studies that polyphenol compounds found in plants can protect against UVB damage and stimulate the immune system [14,15]. Studies have reported that natural chemopreventive compounds such as polyphenols (grape seed extract and ellagic) [52,53], green tea extract [54], and curcumin [55] have been shown to decrease inflammation.

Moreover, using polyphenols in combination with sunscreens or skincare lotions presents a promising approach to effectively counteracting the detrimental effects of UV radiation, thereby protecting the skin against diverse skin disorders arising from excessive sun exposure. Presently, there is a growing interest in investigating the photoprotective and antioxidant capabilities of bioactive compounds within polyphenols, such as rutin, rosmarinic acid, and bilberries (*Vaccinium myrtillus*). These compounds can enhance the sun protection factor (SPF) value and confer multifunctional attributes to sunscreens against UVB radiation.

Rutin, a citrus flavonoid glycoside derived from plant sources, emerges as a noteworthy photoprotective agent owing to its potent antioxidant properties. In the formulation, rutin demonstrated a 40% increase in antioxidant activity and a substantial 70% improvement in photoprotection [56,57]. This evidence supports its efficacy in enhancing photoprotection, possibly due to its anti-inflammatory activity, even at relatively low concentrations. Rutin notably curtails erythema formation, reinforcing its role in elevating photoprotection. Therefore, rutin emerges as a safe and effective bioactive compound suitable for incorporation into multifunctional sunscreens [56,57].

Another compound under scrutiny for its photoprotective potential is rosmarinic acid (RA). RA, recognized for its antioxidant and anti-inflammatory properties, reduced the expression of IL-6, a cytokine implicated in the early response to UVB radiation. RA also demonstrated a substantial increase in IL-10 levels when applied immediately after irradiation. The topical application of RA emulsion further elucidated its anti-inflammatory efficacy by reducing TNF-α levels [53,56,58]. Evaluation of RA as a photoprotective adjuvant ingredient yielded successful results, manifesting an elevation in a sunscreen system’s in vivo sun protection factor (SPF) [59,60].

Bilberries (*Vaccinium myrtillus*), known for their rich content of hydrophilic phenolic compounds and flavonoids, have been identified as effective UV absorbers and are commercially utilized in sunscreen products [61,62]. Bilberries exhibit significant anti-inflammatory, antioxidant, and anti-DNA-damaging effects. These protective attributes of polyphenols are anticipated to contribute to their anti-photocarcinogenic effects, countering various biochemical processes induced or mediated by solar UV radiation [63].

In addition, UVB irradiation induces multiple signaling pathways in keratinocytes. In general, cytokines are considered crucial mediators of the UVB-induced inflammatory response [18]. During the fermentation of plant-derived preparations, microbial enzymatic machinery degraded long-chain polyphenols to yield small polyphenol compounds, increasing their bioavailability and biofunctionality. Small-chain polyphenols like gallic acids, protocatechuic acid, and catechins are more easily absorbed in the digestive tract than long-chain polyphenols [11,14,15]. In this study, we opted to use PEBP, the natural ferment of blueberry, and OMP (a mixture mimicking the main polyphenol compounds released after fermentation) in vivo to study the pleiotropic effects of these polyphenols and their anti-inflammatory properties in a model of skin UVB-induced inflammation. This study evaluated the effects of topical administration of PEBP and OMP extracts on UVB-induced inflammation and skin damage in BALB/c mice. To elucidate the protective mechanism of PEBP and OMP, we analyzed inflammatory factors and components of the NF-κB signaling pathway in the skin tissues of the mice.

UVB radiation causes a massive infiltration of mast cells in the skin, triggering the inflammatory response [47,64,65]. Mast cells play a crucial role in human defense by releasing cytokines such as IL-10, TNF-α, and IL-6 [65,66]. Our study showed that PEBP and OMP significantly reduced mast cell infiltration, consistent with several studies demonstrating the role of polyphenols in reducing inflammatory mast cells in the skin [64]. This reduction in mast cells might be due to the anti-inflammatory effects of polyphenols, achieved by downregulating pro-inflammatory cytokines [15]. Thus, PEBP and OMP might regulate the inflammatory response to UVB exposure and potentially protect against skin damage.

We also reported the acute effects of UVB exposure on epidermal thickness and neutrophils in the skin. Previous studies have identified epidermal thickness and neutrophil induction as notable signs of photodamaged skin [67,68]. Our data demonstrated that UVB exposure led to a significant increase in epidermal thickness and the number of neutrophil cells. However, treatment with PEBP and OMP protected against the loss of epidermal thickness and reduced the number of neutrophil cells in the skin of BALB/c mice. Comparative analysis of non-fermented blueberry juice (NBJ), PEBP, and OMP revealed notable differences in the populations of mast cells and neutrophil cells. Specifically, NBJ-treated mice exhibited higher counts of mast cells and neutrophil cells compared to those treated with PEBP and OMP. This difference is attributed to biotransformations induced by fermentation and catabolic breakdown, which have been proposed to augment the bioavailability of PEBP and OMP [11,14]. Together, these findings suggested that PEBP and OMP might have a protective effect against UVB-induced skin damage, potentially through their regulation of inflammatory responses, restoration of skin thickness, and decreased skin infiltration of neutrophil cells.

To further shed light on the mechanisms underlining the protective effects of PEBP and OMP against UVB damage, epigenome studies were conducted. Recently, there has been a growing interest in prevention as well as therapy using epigenetic modifications. Most human diseases include dysregulated microRNA; therefore, altering their expression provides new opportunities for therapeutic development [69]. Studies have reported that abnormal expression of miRNAs is closely related to the initiation of the differentiation process of epithelial cells such as keratinocytes, inflammation, and carcinogens in the skin [20,22,70,71].

In this study, it was shown that PEBP and OMP stimulated miR-200c and miR-205 expression. In contrast, topical administration of UVB-exposed mice with PEBP and OMP significantly reduced the relative expression levels of miR-155, miR-210, and miR-146a. A study using topical application of baicalin in mice’s skin after UVB showed a significant decrease in miR-146a expression [72]. Furthermore, PEBP has been shown to influence miRNA expression patterns in breast cancer stemness. In particular, it reduces the expression of miR-210 both in vitro and in vivo [14,73]. Upregulation of miR-210 and miR-155 is involved in distant metastasis and inflammatory pathways by increasing NF-κB activation and the pro-inflammatory cytokines TNF-α and IL-6 [24,74]. A study demonstrated that topical application of fisetin significantly inhibited UVB-induced hyperplasia and the infiltration of inflammatory cytokines such as TNFα, IL-1β, and IL-6 [75]. These markers are associated with dysregulation of miR-146a, miR-210, and miR-155 in skin damage and melanoma cancer [21,70,76]. Down-regulation of miR-205 and miR-200c has been related to inflammation conditions such as some cancer, which is correlated with photocarcinogen [74,77]. Korpal et al. found that the miR-200 family impacts EMT and cancer cell migration by directly targeting E-cadherin through transcriptional repressors ZEB1 and ZEB2 [78]. In vitro, studies showed that PEBP could enhance the expression of the tumor suppressor miR-200b in skin cancer cells, which may suggest a role for PEBP in regulating the growth of skin cancer cells [11]. Furthermore, NF-kB activation is one of the main signaling pathways UVB exposure activates [30,79]. Thus, miR-200c and miR-205 are pivotal in regulating melanoma malignancy through interaction with NF-κB [80]. In this study, results illustrated the preventive effect of PEBP and OMP by significantly increasing the expression of miR-200c and miR-205 while decreasing the expression of miR-210, miR-146a, and miR-155, leading to regulation of target gene expression after UVB stimulation. These miRNAs were predicted to be involved in distant metastasis and inflammatory pathways [24,81] and ([82] (p. 25)). According to these results, topical application of PEBP and OMP to the skin exposed to UVB led to regulation of miRNA expression.

Excessive UVB radiation induces skin cells to undergo apoptosis, ultimately leading to skin inflammation [83].The NF-κB signaling pathway plays an essential role in the transcriptional regulation of various genes involved in cell growth, survival, proliferation, apoptosis, adhesion, migration, carcinogenesis, and inflammation [84]. Studies reported that UVB radiation activates NF-κB in BALB/c mouse skin at post-irradiation [28]. NF-κB is involved in the transcriptional activation of the expression of COX-2 and the proinflammatory cytokines TNF-α, IL-1β, and IL-6 [28,85] and the anti-inflammatory cytokines such as IL-10 [86]. These cytokines effectively stimulate neutrophils to migrate to the inflammatory site and play a significant role in UVB-induced skin inflammation [65]. Therefore, we studied the inflammation panel of the cytokine array and found that PEBP and OMP significantly reduced UVB-induced TNF-α, IL-1β, and IL-6 expression levels in mouse skin exposed to UVB. In addition, the present study also demonstrates that PEBP and OMP inhibited the expression of Cox-2, an inflammatory mediator strongly implicated in the process of photocarcinogenesis. Moreover, our data indicate that PEBP and OMP markedly inhibit UVB-induced NF-κB/p65 activation through the modulation of inflammation markers. Therefore, these results suggest that PEBP and OMP offer protection against UVB-induced inflammation in vivo.

We also investigated the effects of all treatment groups, with and without UVB exposure, on DNA methylation. DNA methylation is one of the mechanisms in epigenetics that has gained a lot of attention due to the convincing evidence showing abnormal methylation in cancer cells [87]. However, there is limited knowledge about how DNA methylation changes in correlation with molecular pathways in human skin when exposed to harmful external agents such as UV radiation [66,87]. Results showed that there are extensive DNA methylation changes in skin samples when comparing UVB-exposed to non-exposed skin tissue. These differentially methylated regions (DMRs) were associated with genes reported to be involved in different stages of UVB-induced inflammation, including *Taok1* and *Map4k4*, which are regulated by the MAPK signaling pathway and the NF-κB signaling pathway. These two pathways have a reciprocal relationship, and they can positively regulate each other [88,89]. Topical application of PEBP prior to UVB irradiation might cause the silencing of downstream genes such as *Taok1* and *Map4k4*, leading to blocked activation of the MAPK signaling pathway and subsequent inhibition of NF-kB. Preventing UVB-induced hypomethylation of the *Taok1* and *Map4k4* genes by PEBP may contribute to reducing or preventing skin damage from acute UVB exposure. Furthermore, after a topical application of PEBP, the result showed that *Ifna2* and *Ifnk* were hypermethylated, which might prevent the activation of the Janus kinase (JAK)—signal transducer and activator of transcription (STAT (JAK/STAT)) pathway. Furthermore, IFNs are secreted by damaged cells after UVB irradiation, which leads to activation of the JAK/STAT pathway through binding to downstream genes such as *Ifna2* and *Ifnk* genes, causing inflammation in the skin [90]. On the other hand, the *Runx1* and *Foxp1* genes were found to modulate the effect of the pro-inflammatory cytokine IL-1β, the transcription factor TGF-β, and NF-kB. Interestingly, deletion of *Foxp1* and *Runx1* led to the upregulation of genes linked to inflammasome activation, suggesting their role in regulating the immune response to inflammation [91,92,93]. Therefore, our results showed that PEBP could cause hypomethylation of *Runx1* and *Foxp1*, which has been demonstrated to suppress inflammation through modulating the activation of the NF-kB pathway [91].

Although UVB exposure is the primary trigger of skin cancer, it is unclear if UVB exposure might affect the epigenome as cancer progresses. Early studies on human keratinocytes using the microarray technique revealed that UVB irradiation does not directly generate detectable changes in DNA methylation [94]. Conversely, new research using sequencing-based techniques in mouse skin cancer models found distinct DNA hypermethylation patterns in epidermal skin exposed to UVB and skin cancers caused by UVB [95]. Collectively, these studies shed light on how DNA methylation affects UVB-induced inflammation in skin tissue. It demonstrates the potential contribution of enhanced inflammatory cells driven by UVB to the epigenetic alterations observed in the skin [96]. Polyphenol preparation plays a role in the crosstalk between UVB and other stimuli, signaling through several pathways, such as NF-kB. Our results point towards the key genes and pathways that may contribute to the short-term exposure to UVB.

Overall, this study indicates that PEBP is a highly effective treatment approach for preventing photodamage and skin damage after UVB exposure. Therefore, PEBP holds great potential for preventing the damaging effects of UVB radiation on the skin.

## Figures and Tables

**Figure 1 antioxidants-13-00025-f001:**
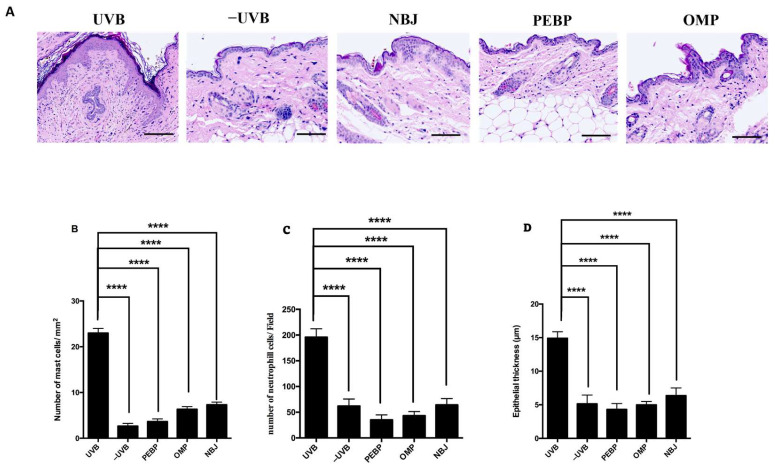
Inhibition of UVB-induced histological alteration by PEBP and OMP in BALB/c mice. (**A**) H&E staining was performed on skin samples 90 min after UVB exposure. Skin non-exposed to UVB (−UVB) mice were treated without UVB. The horizontal line represents 100 μm. The UVB mice exhibited more inflammatory changes, such as increased (**B**) mast cells, (**C**) neutrophil cell count, and (**D**) skin thickness. In contrast, the PEBP and OMP-treated mice demonstrated decreased skin thickness, mast cells, and neutrophil cell count. Original magnification: ×40. **** *p* < 0.0001.

**Figure 2 antioxidants-13-00025-f002:**
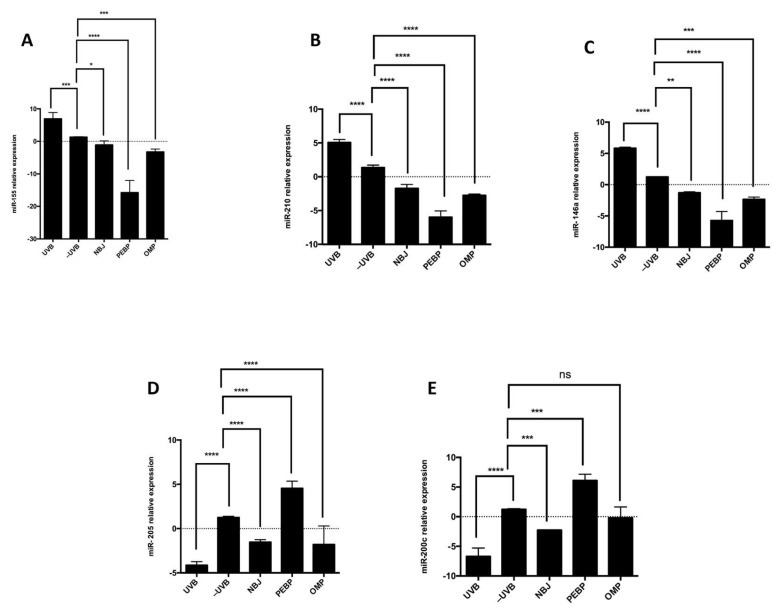
Effect of different treatments on the expression of (**A**) miR-155, (**B**) miR-210, (**C**) miR-146a, (**D**) miR-205, (**E**) and miR-200c after short-term UVB exposure. Expression levels of these miRNAs were measured by RT-qPCR. miR-210, miR-146a, and miR-155 were significantly downregulated (*p* < 0.0001), while miR-200c and miR-205 were upregulated (*p* < 0.0001) compared to non-irradiation control group. The data are the mean ± SEM of at least three independent experiments performed. ns: non-significant, * *p* < 0.05, ** *p* < 0.01, *** *p* < 0.001, **** *p* < 0.0001.

**Figure 3 antioxidants-13-00025-f003:**
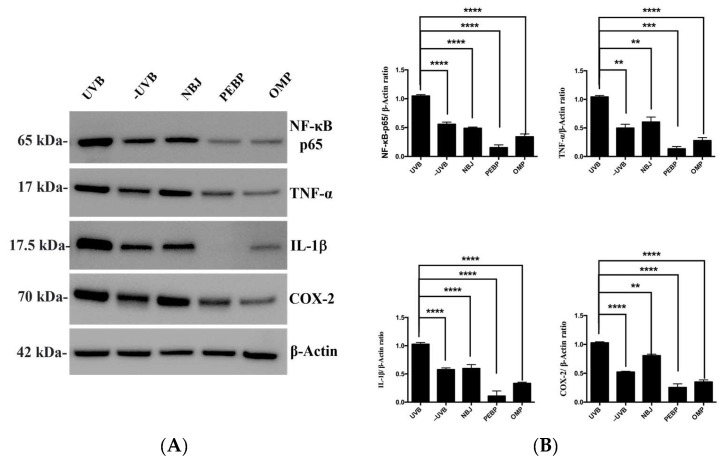
The expression levels of NF-κB-p65, TNF-α, IL-1β, and COX-2 were detected by Western blotting, and the relative intensity was calculated by dividing the intensity of the protein band by that of the control sample on the same blot and then normalizing against the intensity of β-actin on the same membrane. (**A**) Representative Western blot images; and (**B**) Representative quantify and normalize the protein levels using β-actin as the loading control. Values are shown as the mean ± SEM of at least three independent experiments.** *p* < 0.01, *** *p* < 0.001, **** *p* < 0.0001.

**Figure 4 antioxidants-13-00025-f004:**
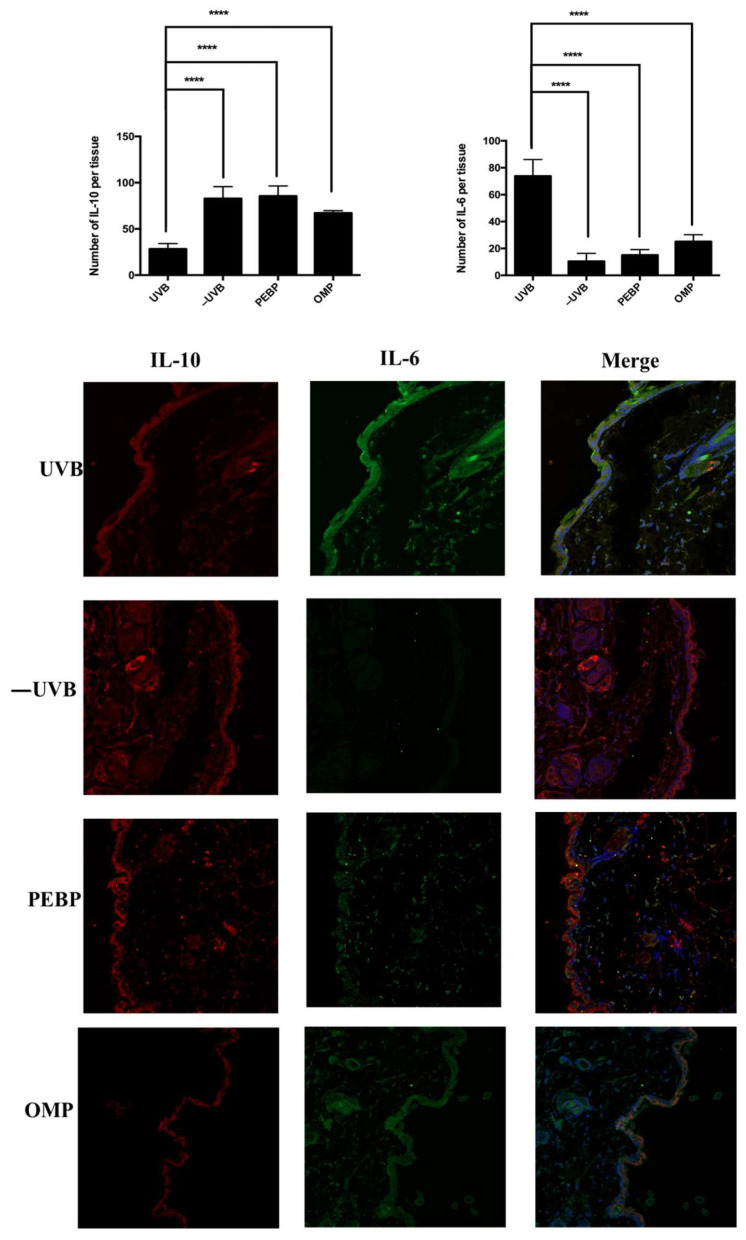
Immunohistochemical analysis skin sample with topical administration of PEBP and OMP after UVB exposure in BALB/c mouse tissues. Tissue sections from skin mice were fixed with 10% formalin, paraffin-embedded, and sectioned into 4-µm-thick slices. Staining was performed using the indicated primary antibodies (IL-6 and IL-10), followed by incubation with fluorophore-conjugated secondary antibodies. Immunofluorescence staining shows the presence of IL-6 (green) and IL-10 (red). Nuclei were stained with DAPI (blue). Quantification was generated from eight fields of view from a representative experiment by confocal microscopy (magnification, 20×). **** *p* < 0.0001.

**Figure 5 antioxidants-13-00025-f005:**
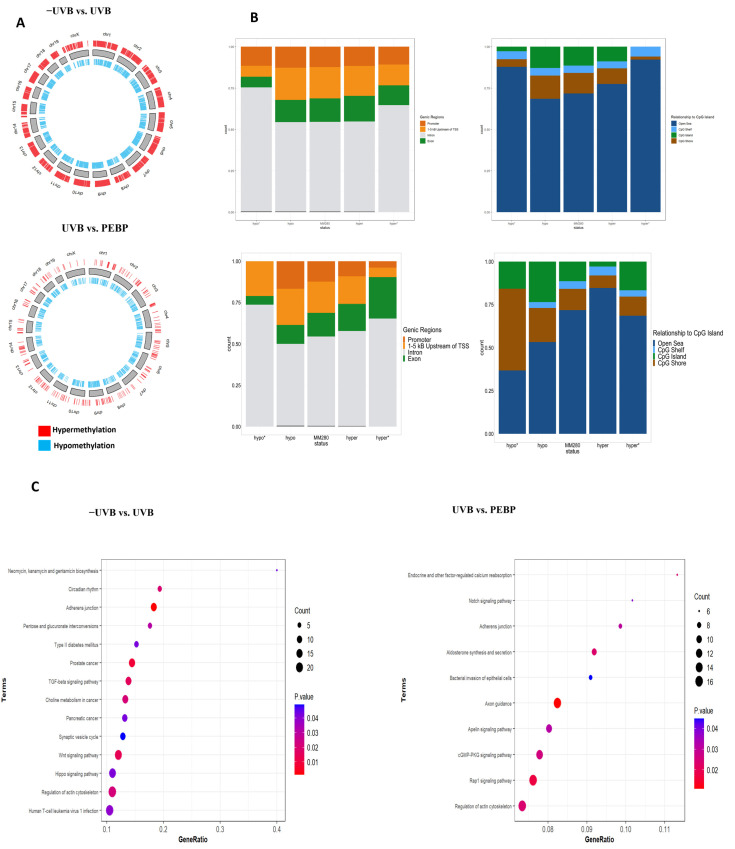
Alterations of DNA methylation following exposure to UVB-irradiation in mouse skin tissues. (**A**) Circos Plot illustrating the distribution of the hyper- and hypo-DMRs relative to their chromosomal location when comparing −UVB to UVB groups (**top panel**) and UVB to UVB + PEBP groups (**bottom panel**). Red bars represent hypermethylated DMRs, while blue bars represent hypomethylated DMRs. Stacked barcharts displaying the distribution of hyper- and hypomethylated DMPs relative to the MM280 array when classified by (**B**) genic regions and (**C**) relationship to CpG island. (**B**) The bars labeled “hypo*” and “hyper*” represent significantly hypo- and hypermethylated DMRs, respectively, while the bars labeled “hypo” and “hyper” represent all hypo- or hypermethylated DMRs identified in the dataset. The bar labeled “MM280” illustrates the overall distribution of probes in the Illumina Infinium Mouse Methylation BeadChip. (**C**) Genomes (KEGG) pathway enrichments of the differentially expressed genes (DEGs) between UVB, non-exposed skin, and PEBP groups. The size and color of the dots represent the gene number and the range of p-values, respectively. (**D**) Boxplots illustrating the differences in DNA methylation levels between the control (−UVB), UVB, and UVB + PEBP groups for selected genes. The resulting box plot displays mean beta values for each sample as data points.

## Data Availability

All of the data is contained within the article.

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
