# Peer review of "Protective Mechanisms of Polyphenol-Enriched Blueberry Preparation in Preventing Inflammation in the Skin against UVB-Induced Damage in an Animal Model"

_antioxidants, 2023, doi:10.3390/antiox13010025_

Round 1
Reviewer 1 Report
Comments and Suggestions for Authors
Dear Authors,
I write you in regard to your manuscript "Protective Mechanisms of Polyphenol-Enriched Blueberry Preparation in Preventing Inflammation in the Skin against UVB-Induced Damage in an Animal Model". This material described interesting findings about a natural topical product, however, authors could have discussed or even presented literature that already used natural compounds against UVB damage in humans, like the rutin (in vivo SPF), V. myritulls (in vivo SPF), rosmarinic acid (comet assay, anti-inflammatory test, in vivo SPF) etc. Certainly, this strategy will put such findings in a scenario parallel to those using humans as the final beneficial targets.
Authors investigated mechanisms of protection against UVB radiation of a polyphenol-enriched blueberry preparation in animal model.
The search of effective natural compounds to protect the skin from UVB radiation is relevant to considering future development of sunscreens. The polyphenol-enriched blueberry was extensively investigated and several markers were determined to observe the protection against UVB rays.
The limitation would be the use of animal model.
The conclusions are consistent with the evidence and arguments presented and do the authors address the main question posed.
I suggested the authors to search the following subjects in references to update and improve discussion: “...authors could have discussed or even presented literature that already used natural compounds against UVB damage in humans, like rutin (in vivo SPF), V. myritulls (in vivo SPF), rosmarinic acid (comet assay, anti-inflammatory test, in vivo SPF) etc”.
The tables and figures were suitable for this manuscript.
Author Response
Dear Reviewer,
Thank you very much for taking the time to review this manuscript. Your constructive comments will help us to improve the quality of our manuscript. Please find the detailed responses below and the corresponding revisions and corrections highlighted in the re-submitted files.
Kind Regards,
Nawal Alsadi

Reviewer 2 Report
Comments and Suggestions for Authors
This paper described the protective effects of Polyphenol-Enriched Blueberry Preparationon on UVB-induced skin damage. Here are my comments:
1: Spell out PEBP in the abstract
2: Line 115 and 134, add the sex of the mice.
3: Line 148: 4mm section, was it supposed to be a 4um section?
4: Line 203: why and what kind of tumor tissues were used for DNA extraction? Was actually 15-20 grams of tissue used? It seemed a lot of tissue was used.
5: Give more details on how to prepare PEBP, NBJ, and OMP.
6: Fig. 1: How were mast cells and neutrophils identified in HE sections? Mast cells and neutrophils need to use specific staining to identify them, for example, toluidine blue for mast cells, and Giemsa Stain for neutrophils. Suggest adding macrophages.
7: Fig. 2: There were no rationales for why those miRNAs were chosen to be examined. It seemed they came from nowhere.
8: NBJ seemed to have significant effects as PEBP and OMP, especially on mast cells and neutrophils shown in Fig. 1. A discussion regarding the difference between NBJ and PEBP should be included.
Author Response

(The authors gave the same response as above.)

Round 2
Reviewer 2 Report
Comments and Suggestions for Authors
The authors improved the manuscript. However, I still think one does not need 15-20 grams of tissue to extract DNA. 15-20g is the weight of an adult mouse. It is unthinkable to use that much tissue for DNA extraction. Please check it with the person who did the experiment and provide references or protocol if you think you did use 15-20 grams of tissue (line 224). In addition, you may be able to identify neutrophils in the tissue stained with Hematoxylin and eosin. It is almost impossible to identify mast cells. Mast cells need a special staining method.
Author Response
Dear Reviewer,
Thank you very much for taking the time to review this manuscript. Your constructive comments will help us to improve the quality of our manuscript. Please find the detailed responses below and the corresponding revisions and corrections highlighted in the re-submitted files.
Best regards,
Nawal Alsadi
